# Expression of Cytoskeletal Proteins (GFAP, Vimentin), Proapoptotic Protein (Caspase-3) and Protective Protein (S100) in the Epileptic Focus in Adults and Children with Drug-Resistant Temporal Lobe Epilepsy Associated with Focal Cortical Dysplasia

**DOI:** 10.3390/ijms241914490

**Published:** 2023-09-23

**Authors:** Darya Sitovskaya, Yulia Zabrodskaya, Petr Parshakov, Tatyana Sokolova, Dmitry Kudlay, Anna Starshinova, Konstantin Samochernykh

**Affiliations:** 1Polenov Neurosurgical Institute—Branch of Almazov National Medical Research Centre, 197341 St. Petersburg, Russia; zabrjulia@yandex.ru (Y.Z.); igh-lab@rambler.ru (T.S.); starshinova_777@mail.ru (A.S.); neurobaby12@gmail.com (K.S.); 2Department of Pathology with a Course of Forensic Medicine Named after D.D. Lochov, St. Petersburg State Pediatric Medical University, 194100 St. Petersburg, Russia; 3Department of Pathology, Mechnikov North-West State Medical University, 191015 St. Petersburg, Russia; 4International Laboratory of Intangible-Driven Economy, National Research University Higher School of Economics, 614070 Perm, Russia; parshakov.petr@gmail.com; 5Department of Pharmacology, Institute of Pharmacy, I.M. Sechenov First Moscow State Medical University, 119991 Moscow, Russia; d624254@gmail.com; 6NRC Institute of Immunology FMBA of Russia, 115552 Moscow, Russia

**Keywords:** drug-resistant epilepsy, focal cortical dysplasia, immunohistochemistry, GFAP, S100, vimentin, caspase-3, children, adult, apoptosis, neurodegeneration

## Abstract

The European Commission of the International League Against Epilepsy (ILAE) has identified glial mechanisms of seizures and epileptogenesis as top research priorities. The aim of our study was to conduct a comparative analysis of the expression levels of cytoskeletal proteins (glial fibrillar acidic protein (GFAP) and vimentin), protective protein S100, and proapoptotic caspase-3 protein in patients with drug-resistant epilepsy (DRE) associated with focal cortical dysplasia (FCD). We aimed to investigate how the expression levels of these proteins depend on age (both in children and adults), gender, and disease duration, using immunohistochemistry. Nonparametric statistical methods were employed for data analysis. In the epileptic focus area of the cortex and white matter in patients with FCD-associated temporal lobe DRE, a higher level of expression of these proteins was observed. Age and gender differences were found for vimentin and S100. In the early stages of disease development, there was a compensatory sequential increase in the expression of cytoskeletal and protective proteins. In patients with DRE, depending on the disease duration, patterns of development of neurodegeneration were noted, which is accompanied by apoptosis of gliocytes. These results provide insights into epilepsy mechanisms and may contribute to improving diagnostic and treatment approaches.

## 1. Introduction

Temporal lobe epilepsy is one of the most prevalent types of focal epilepsy [1], with an incidence rate of 61.4 per 100,000 people [2]. The etiology of temporal lobe epilepsy is complex, and it is most observed in the youngest age group [2]. In fact, approximately 40% of the 3.5 million people who develop epilepsy each year are under the age of 15 [3,4]. Despite advancements in epilepsy treatments and a wide range of anticonvulsant medications, complete seizure control is not possible in 30% of cases [5]. Children with drug-resistant epilepsy (DRE) often experience developmental delay, while adult patients may exhibit increased cognitive impairment, and mortality rates are twice as high as the general population [6]. Surgical intervention yields positive results in only 60% of cases, provides limited improvement in 20% of cases, and has no effect in the remaining 20% [7,8].

Focal cortical dysplasia (FCD), specifically malformations of the cerebral cortex, is one of the most common causes associated with DRE [9,10]. FCD is characterized by focal abnormalities in cortical cytoarchitecture and is closely linked to DRE, particularly in children and young adults. The International League Against Epilepsy (ILAE) has developed a classification system for FCD lesions based on histological findings from resected brain tissue samples. Type I (FCD I) anomalies involve radial or tangential cortical separation, type II (FCD II) is characterized by the presence of dysmorphic neurons and/or balloon cells, and type III (FCD III) refers to anomalies of the cortical separation combined with another brain lesion. Mild malformations of cortical development (mMCD) are designated separately when referring to white matter lesions [11]. However, studying the mechanisms of neuronal functioning in epilepsy necessitates understanding the mechanisms of glial functioning due to the close interaction between these components [12].

Complex degenerative changes have been observed in the white matter within the epileptic focus zone, reflecting epileptic leukoencephalopathy. These changes include demyelination, rarefaction, and microcystic transformation of the neuropil, as well as neuronal devastation foci and atrophy of the temporal lobe cortex [13]. Reactive gliosis and neuroinflammation, which accompany neuronal damage, are also believed to contribute to the emergence of hyperexcitability foci [14]. Recent research has highlighted the critical role of glial cells (astrocytes, microglia, and oligodendrogliocytes) in various neurodegenerative diseases [15]. Glial cells are no longer seen as passive observers of brain function but are now recognized as important contributors to brain pathophysiology. In fact, astrocyte dysfunction has been proposed as a basis for epilepsy, as both animal models and human samples suggest that astrocytes play roles in hyperarousal, neurotoxicity, seizure spread, and other neurogenic functions [15,16]. Furthermore, oligodendrocytes play an important role in epileptogenesis and the development of degenerative processes in DRE [17]. This recognition has led the European Commission of the International League Against Epilepsy (ILAE) to identify glial and inflammatory mechanisms in seizure development and epileptogenesis as a top research priorities, while calling for the identification of glial targets for the development of more specific antiepileptic drugs [18].

Numerous studies have demonstrated an increase in the expression of cytoskeletal proteins (GFAP, vimentin), protective S100 protein, and proapoptotic proteins caspase-3 in the brain tissue of the epileptic focus zone, including cases of FCD [19,20,21,22]. These proteins have been suggested to play a role in the pathogenesis of epilepsy. However, there are limited data available regarding the expression of these proteins based on the duration of disease history and age characteristics in patients with focal DRE associated with FCD.

Traditionally, studies have focused separately on children and adults, despite the fact that the onset of adult DRE associated with FCD often occurs during childhood. Therefore, the objective of our study was to conduct a comparative analysis of the expression levels of cytoskeletal proteins (GFAP, vimentin), the protective protein S100 and the pro-apoptotic protein caspase-3 in patients with FCD-associated temporal lobe DRE. We aimed to investigate how these protein expression levels vary depending on age (children versus adults), sex, and disease duration. The purpose of this analysis was to clarify the dynamics and roles of these proteins in the pathogenesis of DRE.

## 2. Results

### 2.1. Histological Examination

Histological examination of patients of all age groups revealed various types of focal cortical dysplasia (Figure 1).

### 2.2. Immunohistochemical (IHC) Study

#### 2.2.1. GFAP

IHC reactions with antibodies to GFAP revealed proliferation of astrocytes with the development of astrogliosis (Figure 2) both in the cortex and in the white matter in patients with FCD-associated temporal lobe DRE, when compared with the control group. In adults, the changes were more severe (Figure 2a,e): thickening of astrocyte processes was noted; however, the number of processes is small, and there are numerous intercellular contacts, up to the formation of a glial scar. In children, there is a more pronounced proliferation of astrocyte processes (Figure 2c,g), without thickening of the processes.

#### 2.2.2. S100

IHC with antibodies to S100 revealed a more intense staining of the cytoplasm and nuclei of numerous gliocytes in the group with FCD compared with the control group (Figure 3).

#### 2.2.3. Vimentin

IHC with anti-vimentin antibodies revealed positive cytoplasmic staining of numerous reactive glial cells (in internal controls) in patients with FCD-associated temporal lobe DRE (Figure 4). At the same time, cytoplasmic staining of neurons was also found in the cortex of children (Figure 4c).

#### 2.2.4. Caspase-3

IHC reactions with the proapoptotic marker caspase-3 showed a positive reaction in gliocytes of both the cortex and white matter in patients with FCD-associated temporal lobe DRE, while in the comparison group the reaction was negative (Figure 5).

### 2.3. Morphometric Results and Comparative Statistical Analysis

As a result of the analysis of the obtained data using the Kolmogorov–Smirnov test, the normal distribution was not obtained. The medical history of the disease in adults ranged from 6 to 45 years (21 ± 9.9) years; in children, it ranged from 1 to 15 years (6.41 ± 3.9).

The optical density (symb. units, s.u.) of GFAP-positive cells in the FCD-associated temporal lobe DRE adult group in cortex was 0.009–0.183 versus 0.001–0.087 in the comparison group (Table 1), while in white matter the density was 0.002–0.434 versus 0.025–0.06 in the comparison group. For vimentin, it was 0.01–0.431 in the cortex versus 0.001–0.045 in the comparison group, and in white matter it was 0.025–0.151 versus 0.009–0.031 in the comparison group. Quantification of cells that reacted with S100 antibodies was found in cortex patients with FCD-associated temporal lobe DRE 6–47 versus 7–19 in the comparison group, while in white matter it was 42–155 versus 34–60 in the comparison group. For caspase-3, the number (n) of positive cells in the cortex patients with FCD was 14–39 versus 0–1 in the comparison group, and in white matter it was 1–15 versus 0 in the comparison group. The data in the format of the arithmetic mean and their standard deviation are shown in Table 1. Statistical data processing resulted in a significant difference according to the Mann–Whitney U test (*p* < 0.05).

The optical density of GFAP-positive cells in the FCD-associated temporal lobe DRE in children in cortex was 0.003–0.16 versus 0.001–0.095 in the comparison group (Table 2), while in white matter it was 0.008–0.83 versus 0.008–0.0134 in the comparison group. For vimentin, it was 0–0.754 in the cortex versus 0–0.138 in the comparison group, while in white matter it was 0.009–0.309 versus 0.009–0.261 in the comparison group. Quantification of cells that reacted with S100 antibodies was found in cortex patients with FCD-associated temporal lobe DRE 4–38 versus 5–29 in the comparison group, while in white matter it was 9–109 versus 4–35 in the comparison group. For caspase-3, the number of positive cells in the cortex patients with FCD-associated DRE was 0–46 versus 0–2 in the comparison group, while in the white matter it was 0–51 versus 0–1 in the comparison group. The data in the format of the arithmetic mean and their standard deviation are shown in Table 2. Statistical data processing resulted in a significant difference according to the Mann–Whitney U test (*p* < 0.05).

For adults, the proteins caspase-3, S100, and vimentin show higher expression levels in patients with epilepsy compared to those without epilepsy, while GFAP shows a slight increase. In children, caspase-3 and S100 exhibit higher expression levels in patients with epilepsy, while GFAP and vimentin show minimal differences.

### 2.4. Dependence on Gender

Table 3 provides a comparison of protein expression levels in patients with epilepsy and those without epilepsy, differentiated by gender (male or female). In male patients, caspase-3 exhibits higher expression in epilepsy cases compared to non-epileptic cases, while GFAP shows similar expression levels in both groups. S100 expression is elevated in male epilepsy patients compared to those without epilepsy, whereas vimentin expression remains relatively low in both groups. Among female patients, caspase-3 expression is also notably higher in epilepsy cases compared to non-epileptic cases. Similar to males, GFAP expression remains consistent between the two groups. S100 displays elevated expression in female epilepsy patients compared to those without epilepsy, while vimentin expression remains relatively low in both groups. Overall, the data suggests that caspase-3, S100, and vimentin exhibit differential expression patterns between epilepsy and non-epilepsy patients, while GFAP expression remains relatively consistent regardless of the condition.

### 2.5. Protein Expression and the Duration of the Illness

Table 4 provides information about the correlation between protein expression levels in the cortex and the duration of illness in patients with FCD-associated temporal lobe DRE. It suggests that there is a statistically significant positive correlation between caspase-3 expression and illness duration, and a statistically significant negative correlation between vimentin expression and illness duration. However, there is no statistically significant correlation for GFAP and S100 protein expressions with illness duration in these patients. Correlations between white matter protein expression and disease duration have also been studied. All correlation coefficients are statistically significant. The table shows the protein GFAP exhibits a negative correlation with the duration of illness. The protein S100 shows a positive correlation with the duration of illness, while vimentin demonstrates a negative correlation. Lastly, caspase-3 also shows a negative correlation with the duration of illness.

When analyzing the dependence of the level of protein expression on the duration of the illness (Figure 6), it was found that the peak of vimentin expression both in the cortex and in the white matter occurs at the anamnesis at 5 years, and then it decreases in all the studied areas. The expression level of the S100 protein increased depending on the duration of the disease both in the cortex and in the white matter. The level of caspase-3 expression in the cortex increases with time, while in the white matter it decreases with increasing disease duration. The highest level of GFAP expression in the cortex was observed with a duration of the disease from 4 to 13 years, and then its content in the tissue decreased. At the same time, in the white matter, the peak values of GFAP expression occured during a short period (up to 5 years), then its level steadily decreased.

### 2.6. Correlation Analysis Results

Table 5 presents the correlation coefficients between the expression levels of four proteins: GFAP, vimentin, S100, and caspase-3. The expression of GFAP is positively correlated with vimentin, S100, and caspase-3. The expression of vimentin shows no correlation with caspase-3. The expression of S100 exhibits a positive correlation with caspase-3.

### 2.7. Results of Regression Analysis

We report the results of the regression analysis in this section. We report four tables, which vary by the localization of expression (cortex and white matter) and the type of model (base or with interactions). Each table consists of four regression models, with the expression of GFAP, vimentin, S100, and caspase-3 as a dependent variable.

In Table 6, the regression results for the cortex based on the initial models revealed that patients with FCD-associated temporal lobe DRE exhibited higher expression levels in all four proteins, namely GFAP, vimentin, S100, and caspase-3. Additionally, it was observed that children with epilepsy had significantly lower expression of caspase-3 compared to other age groups. However, no significant variations in protein expression were found based on gender.

Moving on to Table 7, the regression results for the cortex based on the models with interaction were examined. In this table, the researchers aimed to test whether the protein expression in patients with FCD-associated temporal lobe DRE varied by age and gender. The findings indicated that patients with epilepsy showed higher expression levels of GFAP, S100, and caspase-3 proteins. For vimentin, both the coefficient for the protein itself and the interaction term were jointly significant.

Furthermore, it was observed that children with FCD-associated temporal lobe DRE had lower expression levels of S100 and caspase-3 compared to adults.

In Table 8, which presents the regression results for the base models for the white matter, the researchers observed that patients with FCD-associated temporal lobe DRE exhibited higher expression levels of vimentin, S100, and caspase-3 proteins compared to the control group. However, there was no statistically significant difference in GFAP expression between the two groups.

Furthermore, the study explored the impact of age and gender on protein expression. It was found that children with FCD-associated temporal lobe DRE had significantly lower expression levels of caspase-3 and S100 proteins in white matter compared to adults with epilepsy.

Regarding gender, the researchers found that the protein expression of S100 was higher in females compared to males among the patients with FCD-associated temporal lobe DRE. However, no significant gender-related differences were observed for GFAP, vimentin, or caspase-3 expression.

Table 9 presented the regression results for the models with interaction, specifically testing if the protein expression in patients with FCD-associated temporal lobe DRE varied by age and gender in the white matter. We found that patients with epilepsy had higher expression levels of GFAP, S100, and caspase-3 proteins compared to the control group. Additionally, for vimentin, both the coefficient for vimentin itself and the interaction term were jointly significant, indicating that the protein expression of vimentin in FCD may be influenced by both age and gender.

Furthermore, the study revealed that children with epilepsy had higher expression levels of caspase-3 compared to adults with FCD-associated temporal lobe DRE.

Based on our findings, we have observed the following patterns of protein expression in the epileptic foci of patients with FCD-associated temporal lobe DRE, which vary depending on gender, age, and disease duration:**Protein expression:** In all studied areas of the cortex and white matter, a significant increase in protein expression was observed across all age groups.**Disease duration:** Caspase-3 levels in the cortex positively correlate with disease duration, while vimentin levels negatively correlate. In the white matter, S100 shows a positive correlation with disease duration, while caspase-3, GFAP, and vimentin exhibit negative correlations.**Dynamics:** The peak expression of vimentin in the cortex occurs around 5 years from disease onset, while GFAP reaches its peak at 8 years, suggesting early and late stages of the disease. In the white matter, both vimentin and GFAP show peak expression at approximately 5 years from disease onset, followed by a decrease in all studied regions. Caspase-3 levels increase over time in the cortex but decrease in the white matter. S100 expression also increases as the disease progresses.**Correlations:** GFAP and S100 exhibit positive correlations with caspase-3, indicating a potential association between these proteins.**Age differences:** In children with DRE, the expression of caspase-3 and S100 is lower compared to adults in both the cortex and white matter. Vimentin expression shows age-dependent changes.**Gender differences:** Vimentin expression varies based on gender. S100 levels are higher in women, while no significant relationship with sex was observed for the other proteins.

These patterns help to clarify protein dynamics in epileptic foci, providing insights into the pathogenesis of FCD-associated temporal lobe DRE.

## 3. Discussion

Our study revealed that the expression levels of all the proteins examined in the epileptic foci of patients with FCD-associated temporal lobe DRE showed an increase, along with individual changes in parameters depending on gender, age, and disease duration.

Brain mosaicism has been shown to play an important role in the etiology of FCD. In some types of FCD, somatic brain mutations in *MTOR*, *AKT3*, or *SLC35A2*, or germline mutations in *DEPDC5* and *NPRL3*, have been identified, some of which belong to the mTOR pathway [11]. Mutations in these genes have an impact on cell metabolism, growth, and proliferation, as well as the prevention of apoptosis. These modifications may result in alterations in cellular reactions in response to damage in the epileptic focus.

### 3.1. GFAP

Glial fibrillar acidic protein (GFAP) is a cytoskeletal protein found in astroglial cells in both white and gray matter [23]. Astrocytes, which express GFAP, play a crucial role in responding to neuronal injuries by undergoing reactive astrogliosis, characterized by cellular hypertrophy, astrocyte proliferation, and increased GFAP expression. This response ultimately results in the formation of a glial scar, which serves to protect healthy cells from potential harm caused by harmful substances [24,25,26].

Astrocytes have several important functions, including energy delivery to neurons through the astrocyte–neuron lactate shuttle [27,28]. They also modulate Ca^2+^ influx, which influences neuronal activity through the release of gliotransmitters [29].

In our study, we observed a significant increase in GFAP expression in the brain tissue of both pediatric and adult patients with FCD. There were no differences in GFAP expression based on the patient’s sex. Regression analysis revealed that while GFAP levels were significantly elevated in both the cortex and white matter of epilepsy patients across all age groups, there was no statistically significant difference in GFAP expression between children and adults.

The lack of age dependence suggests that the reactive production of GFAP by astrocytes is not specific to a particular age group and supports our hypothesis of a common pathogenesis of FCD-associated temporal lobe DRE in both children and adults.

The glial scar formed because of astrocyte activation can have epileptogenic effects, either directly or indirectly epileptogenic through the subsequent actions of cytokines on astrocytes [30]. Reactive astrocytes alter their normal homeostatic functions, such as potassium ion uptake, ion buffering, calcium signaling, and the uptake of excitatory neurotransmitters [31].

We found a statistically significant negative correlation between GFAP expression and disease duration in the white matter, while there were no statistically significant changes in the cortex, suggesting a limitation of reparative processes within the nervous tissue. Additionally, we observed a positive correlation between GFAP and vimentin, S100, and caspase-3. This finding demonstrates the functional synergism between reactive and protective proteins in response to apoptosis.

### 3.2. S100

S100 proteins are a diverse group of more than 20 proteins that bind to Ca^2+^ and have various functions within both intracellular and extracellular contexts [32]. These proteins exhibit specialization due to their unique patterns of cell-specific expression. S100β, which is specific to the brain, possesses two calcium-binding active sites [33,34]. S100β is involved in increasing intracellular calcium concentration through either calcium-dependent channels or he depletion of calcium stores [35]. Exogenous S100β can promote neuronal survival and neurite outgrowth through paracrine, autocrine, or endocrine signaling, thereby impacting cognitive functions [36]. S100β also plays a role in modulating glial–neuronal interactions to facilitate brain development and synaptic transmission, potentially through a G protein-coupled receptor (GPCR) [35]. Studies have indicated that the S100β protein regulates GFAP activation, tubulin polymerization, and DNA repair [37]. At the molecular level, S100β proteins have been shown to modulate various biological activities, such as calcium homeostasis, protein phosphorylation, cytoskeletal dynamics, cellular energy metabolism, cell growth, regulation of cell proliferation, and inflammation, as well as nerve impulse conduction and transmission [38].

Our results demonstrate a consistent increase in S100β expression within the epileptic focus of patients with DRE throughout the course of the disease. Several studies have indicated that elevated levels of S100β in the cerebrospinal fluid and temporal lobe of epilepsy patients may be attributed to increased production or release by dysfunctional astrocytes [39].

Furthermore, we observed higher S100β protein expression in the white matter of women compared to men among individuals with DRE, highlighting the importance of studying epileptogenesis in relation to gender [40].

Interestingly, the level of S100β expression in the cortex and white matter was found to be lower in children with DRE compared to adults with epilepsy. This suggests that protein expression may vary with age, implying distinct patterns of expression in children with epilepsy.

Additionally, we found a positive correlation between S100β protein expression in the white matter and both disease duration and caspase-3 protein expression.

The increased presence of S100β may potentially have proapoptotic effects, as it has been shown to upregulate nitric oxide (NO) expression, leading to neuronal and glial cell death and potentially participating in the pathogenesis of epilepsy [39]. Previous studies have reported that inhibition of NO can prevent seizures [41]. While increased protein levels in epilepsy can serve as a protective and adaptive cellular response to focal zone damage, prolonged overexpression may also contribute to glioneuronal apoptosis and sustained neuroinflammation [42].

### 3.3. Vimentin

There is a significant body of research focused on developing pathogenetic treatments that take into account the molecular genetic status and expression of vimentin in brain tissues [43]. Vimentin is an intermediate filament protein found in tissues of mesodermal origin. In the normal adult central nervous system, vimentin is expressed in ependymal, endothelial, and meningeal cells, as well as in some subpial astrocytes and white matter astrocytes. In pathological tissue, vimentin expression is increased in certain hypertrophied astrocytes and reactive microglial/macrophage cells [44].

In our study, we observed a significant increase in vimentin expression in brain tissue of both adult and pediatric patients with DRE. We also found that vimentin expression is influenced by age and gender. In children, vimentin-positive neurons were found in addition to vimentin-positive gliocytes. Vimentin serves as a marker of immaturity in nervous tissue, as it is initially expressed by nearly all neuronal progenitors in vivo and then replaced by neurofilaments shortly after immature neurons become postmitotic [45]. We found a statistically significant negative correlation between disease duration and vimentin expression, along with GFAP, in the cerebral cortex and white matter.

Vimentin, along with GFAP, plays a role in regulating the response of astroglia to inflammation by modulating vesicular transport and astrocyte hypertrophy [21,46,47]. Vimentin is also crucial for incorporating into GFAP-containing networks of nestin and synemin, proteins synthesized by astrocytes in response to neurotrauma and necessary for their activation [48]. This suggests that the presence of vimentin is more critical for astrocyte activation than the presence of GFAP. Our data regarding vimentin and GFAP expression levels reflect the dynamic development of the pathological process and the cascade interaction of these proteins.

Astrogliosis, resulting from the activation of vimentin- and GFAP-immunoreactive astrocytes, plays a beneficial role in the acute stages of the response to damage. However, in later stages, it limits regenerative potential, impairs neurogenesis in the damaged area, and promotes scar formation [47,49]. Conversely, in astrogliosis with a deficiency of vimentin and GFAP in reactive astrocytes, improved restoration of the damaged brain tissue is observed [47,49]. Additionally, vimentin is involved in regulating axon growth, myelination, apoptosis, and neuroinflammation [50]. The functions of vimentin, like other cytoplasmic intermediate filaments, are correlated with its ability to interact with cellular components involved in signaling, as well as with kinases that control gene regulatory networks. Research has identified a novel form of vimentin present on the plasma membrane surface or released into the extracellular environment under various physiological and pathological conditions. The regulation of the vimentin promoter is complex and appears to involve a combination of positive and negative regulatory elements [51].

### 3.4. Caspase-3

Caspases are a family of cysteine-dependent aspartate-specific proteases that play a critical role in maintaining cellular and organismal homeostasis. They serve as key mediators of both the inflammatory response and apoptosis, making them attractive targets for therapeutic intervention in metabolic diseases [52].

Caspase-3, in particular, has gained significant attention for its important role in tissue differentiation, regeneration, and nervous system development [53].

In our study, we observed a significant increase in caspase-3 expression in brain tissue of adult and pediatric patients with DRE compared to the control group, with higher expression levels in adults than in children. We also found a statistically significant positive correlation between caspase-3 expression in the cortex and disease duration, while a negative correlation was observed in the white matter. Furthermore, we identified a positive correlation between caspase-3 expression and S100 and GFAP.

Co-expression of caspase-3 and GFAP has been shown to serve as a marker and early indicator of neurodegeneration and atrophy in brain tissue [54,55]. Additionally, a correlation between S100β protein and oxidative stress and apoptosis signaling pathways has been observed [56]. Interestingly, caspase-3 expression in the cortex was significantly lower in children with DRE compared to adults with epilepsy, whereas in the white matter it was higher in children. This suggests age-related differences in caspase-3 expression within the DRE patient cohort. However, we did not find significant differences in protein expression based on gender. These findings indicate that the level of caspase-3 expression may independently contribute to the pathogenesis of epilepsy and can vary with age.

Experimental evidence suggests that caspase-3 is present in neurons of the hippocampus and temporal lobe, with increased expression following epileptic seizures [57,58]. Conversely, caspase-3 expression is less frequently detected in astrocytes of rats [59]. It has been proposed that astrocyte apoptosis is activated during and after neuronal apoptotic events, potentially contributing to neuronal death and epileptogenesis [58,59,60,61]. However, studies have shown that apoptosis predominantly occurs in oligodendrocytes in patients with epilepsy, leading to demyelination processes in the brain tissue within the area of epileptic activity [13,22]. Additionally, the level of caspase-3 expression has been identified as a marker of disease severity in individuals with epilepsy [62].

## 4. Materials and Methods

### 4.1. Study Design and Patients

The study follows a case–control design. Biopsies of 60 patients with FCD-associated temporal lobe DRE were studied. The study included 30 adult patients (16 men and 14 women, average age of the patients 31.3 ± 8.6 years) and 30 children (16 boys and 14 girls, average age of the patients 9 ± 5 years). All patients were treated at Polenov Neurosurgical Institute, Almazov National Medical Research Centre (ANMRC), St. Petersburg, Russia (2007–2022). The pre-surgical stage of diagnostics was carried out according to the algorithm of the standard diagnostic complex for examining DRE patients, including clinical observation, study of neurological, neuropsychological, and mental status, and electrophysiological and neuroimaging investigations. The type of epileptic seizures was established in accordance with the International League Against Epilepsy (2017) classification. The patients with generalized motor tonic–clonic seizures prevailed, and focal aware seizures were much less common.

The study included patients with FCD of the temporal lobe without association with the main morphological substrate (such as hippocampal sclerosis, tumors, and vascular malformations). All patients underwent anterior temporal lobectomy under intraoperative electroencephalographic control. Serial anatomical slices of the frontal lobe biopsy were performed, and these were 3 mm thick. Histological sections stained with hematoxylin and eosin were studied, with the help of which the type of FCD was determined. An IHC study was carried out in the area of structural changes.

Figure 7 illustrates the age distributions of patients with epilepsy and without the condition across different age groups.

The material of the comparison group for histological examination, IHC (cortex and white matter of the temporal lobe) was obtained at autopsy in the first 6 h after death from 30 patients. Of these, 15 adults (10 men and 5 women, average age of the patients 45.1 ± 10 years) died from somatic diseases, such as acute myocardial infarction, gastric ulcer complicated by bleeding, mesenteric thrombosis, or pulmonary embolism. The comparison group also included 15 children (9 boys and 6 girls, average age of the patients 6 ± 4 years) who died from decompensation of heart defects, cystic fibrosis, or ulcerative colitis complicated by bleeding. These patients had no history of neurological disorders.

All biopsy studies were carried out in the Research Laboratory of Pathomorphology of the Nervous System of Polenov Neurosurgical Institute.

Data supporting the results of this study are available upon request from the respective author. The data are not publicly available due to privacy or ethical restrictions.

### 4.2. IHC

We performed IHC studies of 60 patients with FCD to obtain an analysis of GFAP, S100, vimentin and caspase-3 content in brain tissue. Biopsies of temporal lobe were fixed with 10% paraformaldehyde in a 0,1 M sodium phosphate buffer, dehydrated in a standard way and embedded in paraffin. IHC reactions were performed on paraffin-embedded 3–5 μm-thick slices of the brain temporal lobe biopsies according to the standard protocol. The following antibodies from Dako (CA, USA) were used for the study: mouse monoclonal anti-glial fibrillary acidic protein (M0761), 1:100; mouse monoclonal anti-vimentin (M0725), 1:200; rabbit polyclonal anti-S100 (Z0311), 1:400; rabbit polyclonal antibody to active caspase-3 by Merckmillipore, Darmstadt, Germany (PC679), 1:100. The EnVision polymer detection system (Dako, CA, USA) was also used. For visualization, the streptavidin–peroxidase polymer ultrasensitive system and DAB chromogen (Sigma-Aldrich, Darmstadt, Germany) were used. The sections were counterstained with Gill’s hematoxylin and then embedded in Bio Mount HM synthetic embedding medium (BIO-OPTICA, Milano, Italy). Additionally, reactions lacking primary antibodies were carried outto ensure the specificity of the observed staining. Histological analysis and microphotography were performed using a Leica DM2500 M microscope equipped with a DFC320 digital camera and using an IM50 image manager (Leica Microsystems, Wetzlar, Germany).

### 4.3. Statistical Analysis

For IHC, positive staining with antibodies to GFAP and vimentin in sections of the temporal lobes was assessed by calculating the optical density (symb. units) of stained cells relative to the background areas in 10 fields of view at ×400 magnification using the PhotoM 1.21 program. The result of the measurement is a coefficient obtained by dividing the density of positive areas with respect to the background. Quantification of the results of IHC reactions (positive staining of IHC cells in sections of the temporal lobes, number) with antibodies to S100 and caspase-3 was performed by counting stained nuclei (×400) in sections in 10 fields of view (ImageG). There was a normal distribution. Data are presented in the format M ± m (arithmetic mean ± standard error). Statistical analysis was performed using Microsoft Office Excel 2010 (USA), Statistica v. 10. The Gaussianity of the sample was determined using the Kolmogorov–Smirnov test. Differences between two samples with measured attributes were determined using non-parametric statistical methods (for example, the Mann–Whitney U test if at least one of the samples was not normal). When testing hypotheses for all criteria, the critical significance level was taken <0.05.

In this study, we employed a regression analysis to estimate the relationship between protein levels and various factors, taking into account the potential effects of age, gender, child status, and epilepsy. Two groups of equations were estimated to examine the association between these variables and the protein levels. The base model, used in the first group of equations, is defined as follows in Equation (1):*protein_i j_*= *α* + *β_1_* · *age_i_* + *β_2_* · *female_i_* + *β_3_* · *child_i_* + *β_4_* · *epilepsy_i_* + *ε_ij_*(1)
where *i* denotes the patient and *j* represents the protein. We estimate separate regressions for each protein. *Age* is the age of patient, while child*_i_* is a binary variable reflecting age less than 18. Female is a binary indicator for female gender, and *ε* is the error term.

To further explore the potential interactions between child status, gender, and epilepsy (2), we extended the base model to include interaction terms. This allows us to assess whether the effect of epilepsy on protein levels varies based on the age and gender of the patient. The second group of equations is as follows:*protein_ij_* = *α* + *β_1_* · *age_i_* + *β_2_* · *female_i_* + *β_3_* · *child_i_* + *β_4_* · *epilepsy_i_* + *β_5_* · *female_i_* · *epilepsy_i_* + *β_6_* · *child_i_* · *epilepsy_i_* + *ε_ij_*(2)

In addition to the variables present in the base model, the extended model includes the following interaction terms: *female_i_ · epilepsy_i_* and *child_i_ · epilepsy_i_*. These terms allow us to investigate whether the relationship between epilepsy and protein levels differs based on the patient’s gender and age.

By estimating separate regressions for each protein, we can examine the specific associations between the variables of interest and the protein levels, while considering the potential moderating effects of gender and age.

## 5. Conclusions

The findings of our study demonstrate the involvement of GFAP, vimentin, S100, and caspase-3 proteins in epileptogenesis. We observed higher expression levels of these proteins within the epileptic focus areas of the cortex and white matter in patients with FCD-associated temporal lobe DRE. Age differences were identified, with children exhibiting distinct patterns of protein expression. Some gender differences were also noted. In the early stages of disease development, there was a compensatory sequential increase in the expression of cytoskeletal and protective proteins. Depending on the duration of the disease, dynamics in the progression of neurodegeneration in patients with DRE could be observed. These changes may result from exhaustion and altered adaptive responses. The indicated findings offer insights into the mechanisms underlying epilepsy and have the potential to enhance diagnostic and treatment strategies. Further research is needed to fully comprehend the functional significance of these protein biomarkers in epilepsy.

### Limitations of the Study

Due to the small number of patients in the study, we were unable to evaluate the distribution of protein expression based on the type of FCD.Only isolated types of FCD (FCD I and II, and mMCD) were included in the study. The types of FCD associated with the underlying pathological substrate (such as hippocampal sclerosis, tumors, and vascular malformations) were not investigated.The pathology department’s archival materials were examined; however, data on the patient’s neurological status were not always presented in full in the medical history, which, among other things, makes it difficult to assess the outcomes according to the Engel classification and the duration of treatment.The study did not analyze the dependence of changes in protein expression on the type and number of seizures.

## Figures and Tables

**Figure 1 ijms-24-14490-f001:**
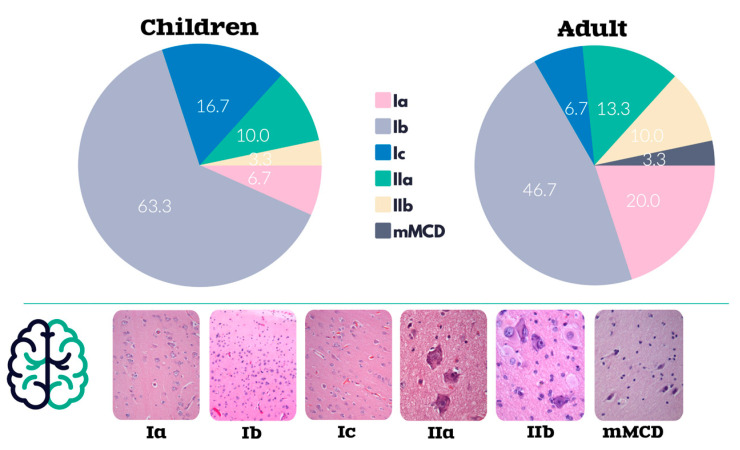
Results of histological examination. Types of FCD in children and adults, %.

**Figure 2 ijms-24-14490-f002:**
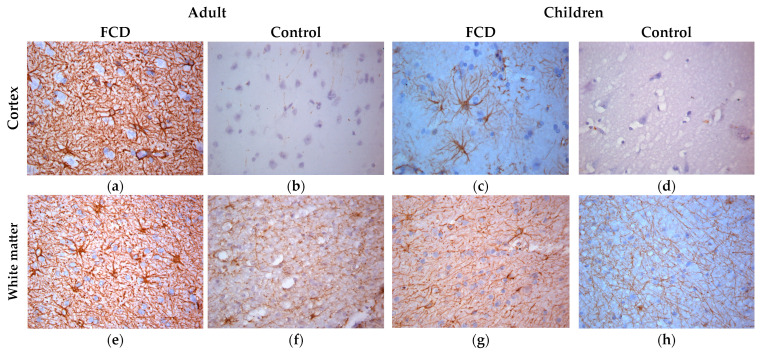
ICH with GFAP, ×400. (**a**,**e**) Adult with FCD; (**b**,**f**) adult control; (**c**,**g**) children with FCD; (**d**,**h**) child control (explanations in the text).

**Figure 3 ijms-24-14490-f003:**
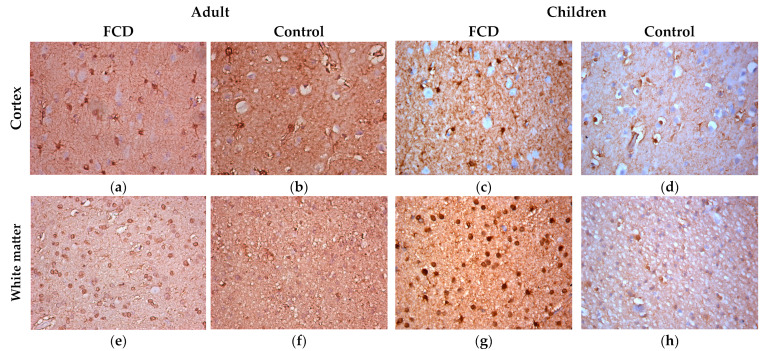
ICH with S100, ×400. (**a**,**e**) Adult with FCD; (**b**,**f**) adult control; (**c**,**g**) children with FCD; (**d**,**h**) child control (explanations in the text).

**Figure 4 ijms-24-14490-f004:**
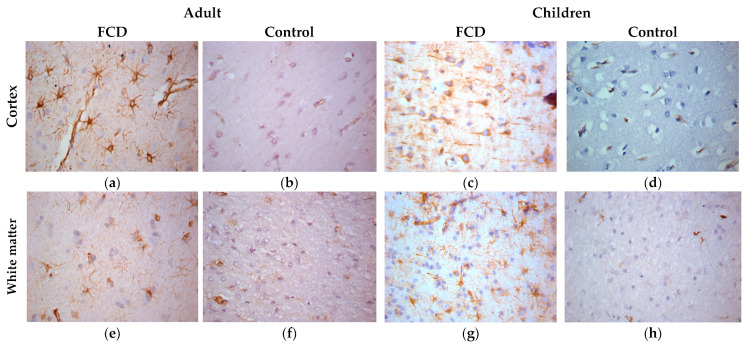
ICH with vimentin, ×400. (**a**,**e**) Adult with FCD; (**b**,**f**) adult control; (**c**,**g**) children with FCD; (**d**,**h**) child control (explanations in the text).

**Figure 5 ijms-24-14490-f005:**
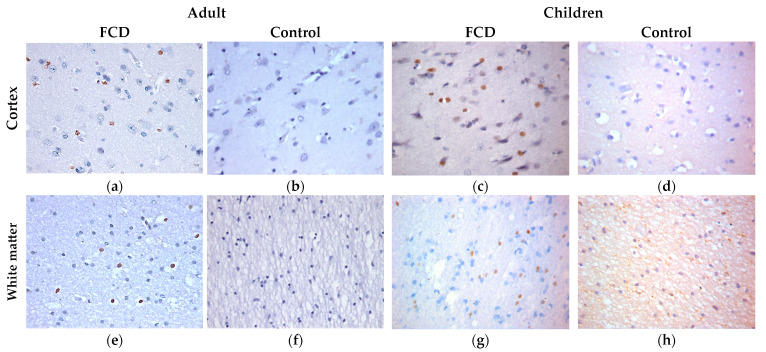
ICH with caspase-3, ×400. (**a**,**e**) Adult with FCD; (**b**,**f**) adult control; (**c**,**g**) children with FCD; (**d**,**h**) child control (explanations in the text).

**Figure 6 ijms-24-14490-f006:**
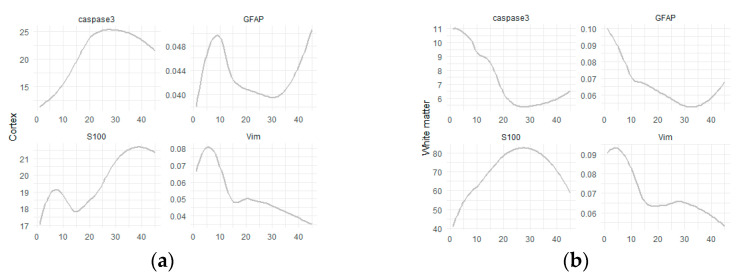
Expression of proteins depending on the duration of the illness. The *x*-axis is the duration of the illness, and the *y*-axis is the level of protein expression. (**a**) Cortex and (**b**) white matter.

**Figure 7 ijms-24-14490-f007:**
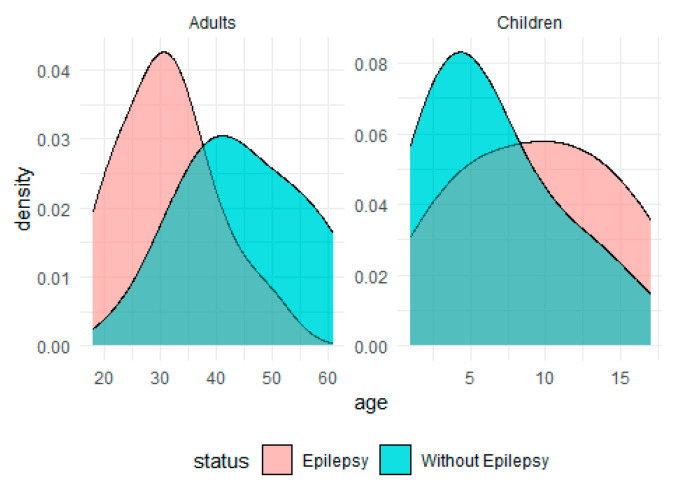
Age distribution of patients with and without epilepsy.

**Table 1 ijms-24-14490-t001:** The level of expression of proteins (symb. units and number) in the cortex and white matter in adult patients with FCD-associated temporal lobe DRE and the comparison group M ± m, *p*-value result (Mann–Whitney U test).

Protein	Cortex		*p*-Value	White Matter		*p*-Value
	FCD Adult	Control Adult		FCD Adult	Control Adult	
GFAP, s.u.	0.042 ± 0.02	0.025 ± 0.006	*p* < 0.05	0.06 ± 0.02	0.038 ± 0.006	*p* < 0.05
S100, n.	19.8 ± 4	11.9 ± 0.7	*p* < 0.001	78.8 ± 16.5	44.8 ± 2.3	*p* < 0.001
Vimentin, s.u.	0.05 ± 0.012	0.2 ± 0.004	*p* < 0.001	0.064 ± 0.016	0.017 ± 0.002	*p* < 0.001
Caspase-3, n.	24.2 ± 2.8	0.033 ± 0.05	*p* < 0.001	5.94 ± 1.9	0 ± 0	*p* < 0.001

**Table 2 ijms-24-14490-t002:** The level of expression of proteins (symb. units and number) in the cortex and white matter in children with FCD-associated temporal lobe DRE and the comparison group M ± m, *p*-value result (Mann–Whitney U test).

Protein	Cortex		*p*-Value	White Matter		*p*-Value
	FCD Children	Control Children		FCD Children	Control Children	
GFAP, s.u.	0.046 ± 0.011	0.035 ± 0.013	*p* < 0.05	0.091 ± 0.05	0.051 ± 0.012	*p* < 0.05
S100, n.	18 ± 5	14 ± 3	*p* < 0.05	50 ± 22	16 ± 5	*p* < 0.05
Vimentin, s.u.	0.077 ± 0.05	0.03 ± 0.016	*p* < 0.05	0.093 ± 0.04	0.058 ± 0.044	*p* < 0.05
Caspase-3, n.	11 ± 6	0 ± 0	*p* < 0.001	12 ± 9	0 ± 0	*p* < 0.001

**Table 3 ijms-24-14490-t003:** Comparison of protein expression levels by gender (M).

Protein	Male		*p*-Value	Female		*p*-Value
	FCD Adult	Control Adult		FCD Children	Control Children	
GFAP	0.04	0.03	*p* < 0.05	0.04	0.03	*p* < 0.05
S100	19.26	12.82	*p* < 0.05	18.53	12.56	*p* < 0.05
Vimentin	0.06	0.02	*p* < 0.05	0.07	0.03	*p* < 0.05
caspase-3	16.77	0.03	*p* < 0.05	18.50	0.08	*p* < 0.05

**Table 4 ijms-24-14490-t004:** Correlation between protein expression in the cortex/white matter and duration of illness.

Protein	Correlation Cortex	*p*-Value	Correlation White Matter	*p*-Value
GFAP	−0.09	0.493	−0.32	0.014
S100	0.19	0.142	0.42	0.001
Vimentin	−0.35	0.006	−0.33	0.011
Caspase-3	0.56	0.000004	−0.29	0.027

**Table 5 ijms-24-14490-t005:** Correlation between protein expression levels.

Protein	GFAP	Vimentin	S100	Caspase-3
GFAP	1.0	0.3	0.3	0.2
Vimentin	0.3	1.0	0.4	0.2
S100	0.3	0.4	1.0	0.5
Caspase-3	0.2	0.2	0.5	1.0

**Table 6 ijms-24-14490-t006:** Regression results for cortex for the base models.

	Dependent Variable:			
	Cortex			
	GFAP	Vimentin	S100	Caspase-3
	(1)	(2)	(3)	(4)
Epilepsy	0.01 ***	0.04 ***	6.14 ***	16.16 ***
	(0.003)	(0.01)	(0.95)	(1.10)
Age	0.0000	0.0000	−0.02	−0.23 ***
	(0.0002)	(0.0004)	(0.05)	(0.06)
Female	−0.001	0.01	−0.56	1.49
	(0.003)	(0.01)	(0.86)	(1.00)
Child	0.01	0.02 *	−1.12	−15.09 ***
	(0.01)	(0.01)	(1.67)	(1.93)
Constant	0.03 ***	0.01	14.01 ***	12.87 ***
	(0.01)	(0.02)	(2.25)	(2.60)
Observations	90	90	90	90
*R* ^2^	0.23	0.37	0.36	0.82
Adjusted *R*^2^	0.19	0.34	0.33	0.81
Residual std. error (df = 85)	0.01	0.03	4.03	4.66
F statistic (df = 4; 85)	6.24 ***	12.31 ***	12.09 ***	95.17 ***

Note: *—*p* < 0.1; ***—*p* < 0.01.

**Table 7 ijms-24-14490-t007:** Regression results for cortex for the models with interaction.

	Dependent Variable:			
	Cortex			
	GFAP	Vimentin	S100	Caspase-3
	(1)	(2)	(3)	(4)
Epilepsy	0.02 ***	0.02 *	8.65 ***	22.65 ***
	(0.01)	(0.01)	(1.67)	(1.63)
Female	−0.005	0.002	−0.37	0.05
	(0.01)	(0.01)	(1.51)	(1.47)
Child	0.02	−0.003	3.18	−1.99
	(0.01)	(0.02)	(2.72)	(2.65)
Age	0.0001	−0.0003	0.04	−0.05
	(0.0002)	(0.0004)	(0.06)	(0.06)
Epilepsy: female	0.005	0.01	−0.39	1.73
	(0.01)	(0.01)	(1.83)	(1.78)
Epilepsy: child	−0.01	0.02	−4.05 *	−12.43 ***
	(0.01)	(0.01)	(2.04)	(1.99)
Constant	0.02 **	0.03	10.30 ***	2.37
	(0.01)	(0.02)	(2.91)	(2.84)
Observations	90	90	90	90
*R* ^2^	0.24	0.39	0.39	0.88
Adjusted *R*^2^	0.19	0.35	0.35	0.87
Residual std. error (df = 85)	0.01	0.03	3.98	3.88
F statistic (df = 4; 85)	4.45 ***	8.83 ***	8.91 ***	98.22 ***

Note: *—*p* < 0.1; **—*p* < 0.05; ***—*p* < 0.01.

**Table 8 ijms-24-14490-t008:** Regression results for white matter for the base models.

	Dependent Variable:			
	White Matter			
	GFAP	Vimentin	S100	Caspase-3
	(1)	(2)	(3)	(4)
Epilepsy	0.03 ***	0.04 ***	33.21 ***	9.24 ***
	(0.01)	(0.01)	(3.74)	(1.24)
Age	−0.0000	−0.0000	−0.01	0.09
	(0.0004)	(0.0004)	(0.21)	(0.07)
Female	0.01	0.001	7.83 **	−0.22
	(0.01)	(0.01)	(3.39)	(1.13)
Child	0.02 *	0.03 **	−29.38 ***	6.15 ***
	(0.01)	(0.01)	(6.58)	(2.19)
Constant	0.03	0.02	42.50 ***	−5.16 *
	(0.02)	(0.02)	(8.86)	(2.94)
Observations	90	90	90	90
*R* ^2^	0.28	0.42	0.67	0.45
Adjusted *R*^2^	0.25	0.39	0.65	0.42
Residual std. error (df = 85)	0.03	0.03	15.88	5.27
F statistic (df = 4; 85)	8.47 ***	15.09 ***	42.96 ***	17.13 ***

Note: *—*p* < 0.1; **—*p* < 0.05; ***—*p* < 0.01.

**Table 9 ijms-24-14490-t009:** Regression results for white matter for the models with interaction.

	Dependent variable:			
	White Matter			
	GFAP	Vimentin	S100	Caspase-3
	(1)	(2)	(3)	(4)
Epilepsy	0.004	0.04 ***	27.08 ***	6.11 ***
	(0.01)	(0.01)	(6.61)	(2.18)
Female	−0.003	−0.01	−1.11	0.004
	(0.01)	(0.01)	(5.98)	(1.98)
Child	−0.01	0.05 **	−31.14 ***	0.31
	(0.02)	(0.02)	(10.76)	(3.55)
Age	−0.0005	0.0002	−0.05	0.01
	(0.0005)	(0.0005)	(0.23)	(0.08)
Epilepsy: female	0.02	0.01	13.16 *	−0.16
	(0.01)	(0.01)	(7.24)	(2.39)
Epilepsy: child	0.03 *	−0.02	1.33	5.52 **
	(0.02)	(0.02)	(8.08)	(2.67)
Constant	0.06 **	0.01	47.66 ***	−0.31
	(0.02)	(0.02)	(11.52)	(3.81)
Observations	90	90	90	90
*R* ^2^	0.32	0.43	0.68	0.47
Adjusted *R*^2^	0.28	0.39	0.66	0.44
Residual std. error (df = 85)	0.03	0.03	15.75	5.20
F statistic (df = 4; 85)	6.65 ***	10.36 ***	29.65 ***	12.44 ***

Note: *—*p* < 0.1; **—*p* < 0.05; ***—*p* < 0.01.

## Data Availability

The data presented in this study are available on request from the corresponding author. The data are not publicly available due to privacy reasons.

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
