# Peer review of "Expression of Cytoskeletal Proteins (GFAP, Vimentin), Proapoptotic Protein (Caspase-3) and Protective Protein (S100) in the Epileptic Focus in Adults and Children with Drug-Resistant Temporal Lobe Epilepsy Associated with Focal Cortical Dysplasia"

_ijms, 2023, doi:10.3390/ijms241914490_

Round 1
Reviewer 1 Report
This neuropathologic study was aimed to perform a comparative analysis of the expression levels of cytoskeletal proteins, (GFAP, vimentin), protective protein S100, and proapoptotic Caspase-3 protein in patients with drug-resistant epilepsy (DRE) due to a focal cortical dysplasia (FCD). In the epileptic focus area of the cortex and white matter, in patients with FCD-associated DRE, a higher level of expression of these proteins was found. Age and gender differences were found for Vimentin and S100. In the early stages of disease development, there was a compensatory sequential increase in the expression of cytoskeletal and protective proteins. Depending on the disease duration, dynamic patterns of neurodegeneration were observed, accompanied by apoptosis of gliocytes.
Clinical and neuropathological data are clearly reported and discussed. Statistical analysis is adequate.
I would indicate some minor points that should be addressed by the Authors:
1) In the Abstract, the sentence " In the epileptic focus area of the cortex and white matter in patients with FCD-associated DRE; a higher level of expression of these proteins was observed" should be rewritten, substituting the semicolon with a comma: " In the epileptic focus area of the cortex and white matt er in patients with FCD-associated DRE, a higher level of expression of these proteins was observed".
2) Again in the Abstract, in the statement "Depending on the disease duration; patterns of dynamics in neurodegeneration development; accompanied by apoptosis of gliocytes; could be observed in patients with DRE" the semicolons should substituted by commas. Furthermore, the sense of "patterns of dynamics in neurodegeneration development" is not clear. Perhaps, the entire sentence should be rewritten in a more understandable way.
3) In the Introduction, page 2, line 58, "Complex of degenerative changes have been observed..." should be "Complex degenerative changes have been observed...".
Author Response
Dear Reviewer!
On behalf of the team of authors, I sincerely thank you for your time and your helpful comments.
In response, we are sending you the results of the corrections.
1. Corrected in the abstract.
2. Corrected in the abstract.
3. Corrected in the introduction.
Thank you again.
Sincerely, Daria Sitovskaya.
Reviewer 2 Report
Authors studied expression of cytoskeletal proteins (GFAP, vimentin), proapoptotic protein (caspase-3) and protective protein (S100) in the cortex and white matter of temporal lobe in adults and children with DRE associated with FCD. This manuscript includes clear images of IHC, which can help readers understand the underlying histological pathomechanism of DRE and the difference between adults and children. However, there are some critical and minor points to clarify or ameliorate as follows:
1) Authors should clearly state all patients suffered from temporal lobe epilepsy. Part of Title should be changed from “Drug-Resistant Epilepsy” into “Drug-Resistant Temporal Lobe Epilepsy”, as well as it should be added in the Study design and Patients section.
2) As authors mentioned, there are three major types of FCD such as type I. II, III. Did authors conduct this study separately in terms of FCD types? Authors also should clearly describe that in the context.
3) Did all patients have FCDs in the temporal lobe? And, did all specimens include FCDs? Authors should describe that in the manuscript.
4) More demographics of patients are needed, including information of experienced medicine, treatment duration, seizure frequency, and distribution of Engel classification after surgery.
5) How did author determine epileptic foci? It is always not easy to determine epileptic foci. MRI findings don’t always indicate epileptic foci.
6) In this study, the type of epileptic seizures was established in accordance with ILAE (2017) Classification. Therefore, words explaining seizures should be ameliorated, such as “complex partial seizures”, “secondary generalization”, and “simple partial seizures”.
7) Figure 6 titled “Age distribution of patients with and without epilepsy” is to be changed into Figure 7
8) Limitation of the study should be added in the manuscript.
Author Response
Dear Reviewer!
On behalf of the team of authors, I sincerely thank you for your time and your helpful comments.
In response, we are sending you the results of the corrections.
1. All patients in our study had temporal lobe epilepsy. According to your remark, corrections have been made throughout the text of the article.
2. Unfortunately, it was not possible to conduct a study on the types of FCD due to the extremely small number of patients in some groups of FCD. The data are specified in the text of the article.
3. All patients had FCD in the temporal lobe, lesions of other locations were excluded from the study. In all cases, FCD was detected; cases without FCD were also excluded from the study. The data are specified in the text of the article.
4. In view of the fact that the archival material of the pathology department was studied, medical information was not always presented in full. The data were refined and included within the limitations of the study.
5. The authors identified the epileptic focus through histological and immunohistochemical studies. The data are specified in the text of the article.
6. Corrections have been made in the text of the article
7. The figure caption has been corrected.
8. Limitation of the study has been added to the text of the article.
Thank you again.
Sincerely, Daria Sitovskaya.